# Wine Cork Closures Impacts on Dimethyl Sulfide (DMS) and Precursors (DMSP) Equilibrium of Different Shiraz Wines during Accelerated Bottle Ageing

**Rémi De La Burgade** [1], **Valérie Nolleau** [1], **Teddy Godet** [1], **Nicolas Galy** [2], **Dimitri Tixador** [2], **Christophe Loisel** [2], **Nicolas Sommerer** [1] and **Aurélie Roland** [1,*]

[1] SPO, Université Montpellier, INRAE, Institut Agro, 34060 Montpellier, France
[2] DIAM Bouchage, 3 Rue des Salines, 66400 Céret, France
[*] Correspondence: aurelie.roland@supagro.fr; Tel.: +33-4-99-61-22-98

**Abstract:** Dimethyl sulfide (DMS) is a flavor compound, characteristic of the truffle aroma in red wines, and is well-known to be a fruity exhauster. DMS comes from the degradation of dimethyl sulfide potential (DMSP) during winemaking. Up to now, little is known about the role of the closure on the DMSP degradation during ageing. For that purpose, the effect of four micro-agglomerated wine cork closures was studied on the DMS/DMSP equilibrium, along with six other volatile sulfur compounds (VSC), was investigated in six Shiraz wines. After three months of accelerated bottle ageing, DMS levels increased significantly in all bottles. The most permeable closures induced a lesser accumulation of DMS, suggesting that DMS could be dependent on the redox status of the wine. At the same time, the DMSP decrease was proportional to the permeability of the closures. For the first time, a possible implication of closure permeability on DMSP degradation was observed.

**Keywords:** dimethyl sulfide; dimethyl sulfide potential; micro-agglomerated closure; oxygen transfer rate; Shiraz wine





## 1. Introduction

Dimethyl sulfide (DMS) is a volatile sulfur compound that participates to the bouquet of "late harvest" [1] and dry [2] white wines and enhances its quality even at low concentrations [3]. Indeed, DMS is well-known to raise the fruity note in wines by exhausting esters families especially [4–6]. In red wines, DMS could be perceived as green olive [7] when its concentration was above 100 µg/L for Shiraz and Xinomavro wines. DMS is also described as black olive, truffle and undergrowth for Shiraz and Grenache wines coming from the Rhone Valley and stored for 2 to 7 years [6]. At high concentrations (50 to 100 µg/L), DMS was considered as a faulty sulfurous aroma when studied in a dearomatized Spanish Grenache wine from the 2004 vintage [5]. DMS contents are usually above their perception threshold, which in red wines varies from 27 µg/L [7] to 60 µg/L in Cabernet Sauvignon [4].

DMS has been quantified in all kinds of wines [8]. In white wines, DMS concentrations were between 0 and 474 µg/L [8,9], with higher concentrations in Riesling wines (85–474 µg/L). In rosé wines, the levels of DMS were between 5 to 20 µg/L [8] but few data are available in the literature, preventing the generalization of this range. Concerning red wines, DMS was found in a wide range of concentrations, from 42 to 910 µg/L in Cabernet Sauvignon [4], 18 to 104 µg/L in Portuguese wines [5], 3.4 to 15.6 µg/L in Grenache and between 3.2 to 46 µg/L in Shiraz wines [6].

DMS biogenesis in wine is complex because different aroma precursors are implicated in its metabolism. Indeed, DMS is a varietal aroma, mainly coming from the degradation of *S*-methylmethionine (SMM) present in grapes [6,10], either during alcoholic fermentation or wine ageing. DMS can also be produced during wine fermentation, mostly from cysteine

through the sulfur assimilation pathway [11] or from dimethyl sulfoxide (DMSO) [7]. Recently, it has been reported that after 12 months of wine ageing, 2% of DMS originated from DMSO hydrolysis and 23% from SMM in Shiraz wine [12], with no plateauing at the end.

Because all the DMS precursors in wine have not been fully identified up to now, it was important to use a global indicator to monitor DMS release in wine: the DMS potential (DMSP). Suggested a few years ago, DMSP corresponds to an analytical method based upon alkaline degradation of wine by NaOH and further SPME-GC-MS/MS analysis of released DMS [13]. Up to now, the conversion of DMSP into DMS had been described as only pH-dependent [13] from a chemical point of view, but little is known about its degradation under oenological conditions.

Wine closures played a determinant role in wine quality during bottle ageing. Indeed, the closure impact was twofold, since closures induced variations in oxygen exposure [14], and can have a role in the scalping of flavor compounds [15,16]. In addition, some studies showed that wine closures may have an impact on volatile sulfur compounds' (VSC) formation during ageing [17–19]. Contrary to polyfunctional thiols (3-sulfanylhexan-1-ol and its acetate), whose concentrations decreased during wine ageing [20], both $H_2S$ and MeSH levels could possibly increase, while DMS content systematically rose [13,21]. Nevertheless, many publications only studied a limited number of samples in regard to wine closures, making result generalization difficult.

The aim of this study was to complement the knowledge about the DMS/DMSP equilibrium in wine as well as six other VSC in the function of different closure permeabilities. For that purpose, accelerated ageing conditions were used on six different bottled wines with four microagglomerated closure types. The correlation between closure permeabilities and the concentrations of DMS/DMSP was assessed by GC-MS/MS.

## 2. Materials and Methods

### 2.1. Chemicals

Ethanol (≥99.8%) was purchased from VWR (Rosny-sous-Bois, France). Tartaric acid (≥99.5%), sodium hydroxide (≥98%), magnesium sulfate heptahydrate (≥99%) and reference standards of ethanethiol (EtSH; ≥98.5%), diethyl sulfide (DES; 98%), diethyl disulfide (DEDS; 99%) and thiophene (TP; ≥99%) were supplied by Sigma Aldrich (Saint-Quentin-Fallavier, France). For other reference standards, dimethyl sulfide (DMS; >99%) and dimethyl disulfide (DMDS; ≥98%) were purchased from Fluka (Charlotte, NC, USA), *S*-methyl thioacetate (SMTA; ≥98%) from Alfa Aesar (Kandel, Germany) and *S*-ethyl thioacetate (ETA; 97%) from Lancaster Synthesis (Morecambe, England).

### 2.2. Wines

Shiraz wines were obtained from six different local wineries in France. The name, denomination of origin, vintage year and ageing type are summarized in Table 1. A volume of 2 hL of each wine was collected.

**Table 1.** General information on the six Shiraz wines studied.

| Wine | Denomination of Origin | Vintage Year | Ageing Type |
|------|------------------------|--------------|-------------|
| CR1 | Côtes-du-Rhône | 2020 | Tank |
| CR2 | Côtes-du-Rhône | 2019 | Barrel |
| LR1 | Languedoc-Roussillon | 2019 | Barrel |
| LR2 | Languedoc-Roussillon | 2019 | Tank |
| LR3 | Languedoc-Roussillon | 2019 | Barrel |
| LR4 | Languedoc-Roussillon | 2019 | Barrel |

*2.3. Wine Sampling*

2.3.1. Wine Closures

Four types of microagglomerated cork wine closures were provided by DIAM Bouchage (Céret, France). The dimensions for all closures were 44 × 24 mm. Oxygen transfer rates (OTR) and oxygen initial rates (OIR; this parameter corresponds to the initial concentration of oxygen contained in the closure and released in the bottle headspace before any OTR) were provided by the supplier of wine closures (Table 2).

**Table 2.** Types and permeability parameters for the four micro agglomerated cork closures (OIR: oxygen initial rate; OTR: oxygen transfer rate).

| Closure | Closure 1 | Closure 2 | Closure 3 | Closure 4 |
|---|---|---|---|---|
| OIR (mg $O_2$) | 0.91 ± 0.04 | 1.92 ± 0.21 | 1.98 ± 0.32 | 2.31 ± 0.20 |
| OTR (mg $O_2$/year) | 0.19 ± 0.02 | 1.07 ± 0.29 | 1.15 ± 0.40 | 1.79 ± 0.36 |

2.3.2. Wine Bottling

The wines were bottled in March 2021 at a local manufacturer (Vivelys, France) using conventional winemaking practices. Empty Bordelaise bottles (0.75 L) were placed manually on the filling machine. To decrease the quantity of oxygen introduced into the bottles, empty bottles were first flushed with nitrogen, and, to normalize wine bottles, the first and last ones were not used. Once the bottles were filled with 0.75 L of wine, wine closures were inserted using a manual vacuum corker. For each wine, four closures were used for bottling, using 36 bottles for each closure (closures 1 to 4). Each modality (i.e., one specific wine bottled with one specific closure) was made in triplicate. A total of 144 bottles were filled for the six wines. At the beginning of ageing, 72 bottles were analyzed (3 bottles per type of wine and closure) to characterize the initial concentration of volatiles and oenological parameters.

2.3.3. Accelerated Ageing Mode

Out of 144 bottles, 72 bottles (corresponding to 3 bottles per type of wine and type of closure, $t_3$) were placed in an oven at 35 °C for 3 months. The bottles, placed horizontally, were equally distributed within the oven.

*2.4. Chemical Analysis*

2.4.1. Oenological Parameters

The following oenological parameters were measured for each wine: sugars, alcohol, total acidity, volatile acidity, free $SO_2$, total $SO_2$, pH, malic acid, lactic acid, iron, copper, dissolved $CO_2$, dissolved $O_2$, and active $SO_2$ (Table S1).

2.4.2. Analysis of Volatile Sulfur Compounds by SPME-GC-MS/MS

Sample Preparation

DMS along with 6 other VSC (Table 3) were analyzed using SPME-GC-MS/MS approach. For each sample, $MgSO_4 \times 7H_2O$ (2.5 g) was added to an SPME vial (20 mL) as described by [22]. A volume of 10 mL of the wine sample was put into the vial, then an internal standard solution (thiophene; 75 µg/L; 80 µL) was added. The vial was then sealed with a Teflon-faced septum and stirred by a vortex.

For the calibration samples, a model wine solution was used (tartaric acid (5 g/L), ethanol (12%), pH 3.5).

Analysis by SPME-GC-MS/MS

The VSC were analyzed by SPME-GC-MS/MS technique. The extraction and GC method was adapted from the method developed by Fedrizzi et al. [22]. The sample was stirred at 500 rpm for 5 min at 35 °C. A 50/30 µm DVB/CAR/PDMS Stableflex 2 cm SPME fiber (Supelco, Bellefonte, Saint Quentin Fallavier, France) was used for extraction. The

SPME fiber was inserted into the vial headspace for a 30 min equilibration and was then removed and inserted into the GC injector for 5 min at 240 °C in splitless mode. Chromatographic analysis was performed with a Thermo Trace GC Ultra gas chromatograph coupled with a TSQ 8000 triple quadrupole mass spectrometer (Thermo Scientific, Waltham, MA, USA) equipped with a 30 m × 0.25 mm I.D × 1.00 μm film thickness ZB-WAX fused-silica capillary column (Phenomenex, Le Pecq, France), with a constant helium flow of 0.5 mL/min. The oven temperature program was as follows: 35 °C held for 5 min, heated to 40 °C at a rate of 1 °C/min, held for 1 min, then heated to 240 °C at a rate of 10 °C/min, with a final hold time of 1 min.

**Table 3.** Analytical performances for the 7 VSC analyzed (LOD: limit of detection; LOQ: limit of quantification; $R^2$; linearity; repeatability; accuracy; intermediate reproducibility).

| | LOD | LOQ | $R^2$ | Linearity | | Repeatability (n = 5) | Accuracy (n = 3) | | Intermediate Reproducibility |
|---|---|---|---|---|---|---|---|---|---|
| | μg/L | μg/L | | Min μg/L | Max μg/L | RSD % | Low RSD % | High RSD % | RSD % |
| DMS | 9 | 30 | 0.914 | 60.0 | 480.0 | 17% | 93% | 95% | 18% |
| EtSH | 0.5 | 1.7 | 0.922 | 25.2 | 201.6 | 19% | 95% | 97% | 19% |
| DES | 0.1 | 0.2 | 0.962 | 13.4 | 107.2 | 24% | 100% | 100% | 17% |
| SMTA | 0.03 | 0.1 | 0.979 | 70.0 | 560.0 | 10% | 92% | 96% | 8% |
| ETA | 0.1 | 0.3 | 0.880 | 13.4 | 107.2 | 14% | 90% | 96% | 40% |
| DMDS | 0.007 | 0.02 | 0.941 | 16.3 | 130.4 | 4% | 100% | 100% | 16% |
| DEDS | 0.001 | 0.005 | 0.967 | 15.5 | 124.0 | 4% | 100% | 100% | 12% |

The mass spectrometer was equipped with an electron impact ionization source (EI). MS acquisition was performed in multiple reaction monitoring (MRM) mode. The MS parameters and MRM mode were adapted from the method developed by Slaghenaufi et al. [23]. The optimized transitions and collision energy are reported in Table S2. The transfer line and source temperature were both set at 220 °C.

Analytical Validation

To check the analytical method based on previous methodologies ([22,23]), several parameters were calculated before quantifying the wine samples (Table 3): linearity, accuracy, repeatability, intermediate reproducibility, limit of detection (LOD) and limit of quantification (LOQ).

Linearity was calculated by spiking model wine samples at 7 concentrations for all VSC and an internal standard (thiophene) at a constant concentration. The concentration ratios against the respective area ratios were calculated, and linearity was evaluated through a lack-of-fit test.

Accuracy, repeatability and intermediate reproducibility were measured by spiking samples of Shiraz wine at several levels and were expressed as the recovery and mean RSD (%), respectively. To evaluate accuracy, one Languedoc-Roussillon wine and one Côtes-du-Rhône wine were spiked at 3 different levels (low, medium, high) for each VSC (n = 5). To measure repeatability and intermediate reproducibility, these 2 wines were spiked at 3 different levels (low, medium, high) and measured on 3 different days (n = 5).

The LOD and LOQ were calculated for each compound for a signal to noise (S/N) ratio equal to 3 and 10, respectively (n = 10).

2.4.3. Analysis of DMSP by GC-MS/MS

DMSP analysis (Table S1) was also performed by a GC-MS/MS approach and following the protocol of [6] without modification. Briefly, 10 mL of DMS-free wine (stripping with nitrogen gas) were spiked with deuterated dimethylsulfonium propanoic acid chloride ($[^2H_6]$-DMSPA chloride), then treated under alkaline conditions with pellets of sodium hydroxide and heated at 100 °C for 1 h before analysis by GC-MS/MS.

## 3. Results

### 3.1. Analytical Validation of SPME-GC-MS/MS Analysis

To guarantee reliable results, the analytical method was validated according to the recommendations of the International Organization of Vine and Wine (OIV) [24]. All validation parameters were provided in Table 3.

The first parameter that had to be measured in a validation procedure is the matrix effect. Indeed, this parameter allowed us to evaluate the degree of specificity of the method. It was evaluated by introducing the seven compounds at the same concentrations into three different matrices (a model wine and two Shiraz wines, respectively, from Languedoc-Roussillon and Côtes-du-Rhône). In practice, matrix effect was evaluated by calculating the z-score parameter as reported in the literature [25]. There was no matrix effect for EtSH and DMDS since z-scores were inferior to 2 for both matrices. For the other compounds, z-scores ranged from 2.496 to 7.018, which theoretically entailed the need for calibration in a red wine matrix. For practical aspects, we first checked that measurement accuracy for these compounds was satisfactory with calibration curves obtained with synthetic wine instead of red wine.

In practice, quantification was performed by GC-MS/MS under multiple reaction monitoring mode using thiophene as the internal standard. Calibration curves were obtained in the model wine ranging from about 10 to about 500 µg/L for each analyte by plotting the peak area ratio for quantifier ions ($A_{analyte}/A_{standard}$) versus corresponding concentration ratios. The linearity of these curves was assessed with the lack-of-fit test. The linear model was found adequate for all analytes except for DEDS, for which a quadratic model was most adequate. For each VSC, $R^2$ values were in the range of 0.88 to 0.98 and thus deemed suitable for the pursuit of the analysis.

Accuracy was evaluated by adding known quantities of synthetic VSC into two different wines at two concentration levels (low and high). In parallel, control samples (non-spiked wines) were analyzed to distinguish the natural quantity of VSC from the added one. By comparing the theoretical spiked quantity and calculated quantity, we evaluated the accuracy of the method that ranged from 92% to 100% for all analytes under our conditions, ensuring reliable results. Indeed, according to OIV resolutions, accuracy should range from 80% to 120% to be satisfactory, meaning that the analytical bias is lower than 20%.

Repeatability was studied by using two Shiraz wines at three different concentrations, spiked the same day and calculated as the relative standard deviation (RSD%). RSD ranged from 4% to 24% for all analytes, which was found acceptable. OIV recommended a repeatability value below 20%, which was the case for all the VSC except for DES at 24%. Yet, DES repeatability was considered close enough to OIV resolutions to warrant pursuing the analysis.

The intermediate reproducibility was also calculated as RSD (%) by analyzing two Shiraz wines (n = 5) on three different days using the same machine and user for each experiment. Reproducibility ranged from 8% to 19% for all compounds except for ethyl thioacetate (ETA), which exhibited an RSD close to 40%. To ensure that RSD was below 20%, a calibration curve for ETA was established daily to solve this reproducibility problem.

Limits of detection (LOD) and quantification (LOQ) were calculated on a set of 20 samples using a signal to noise ratio equal to 3 and 10 respectively. For all VSC, LOD was between 0.001 µg/L and 9 µg/L and LOQ between 0.005 µg/L and 30 µg/L allowing us to have a method sufficiently sensitive considering perception thresholds of these compounds.

### 3.2. Characterization of VSC during Accelerated Wine Bottle Ageing

3.2.1. Initial Characterization of Wines

Firstly, the oenological parameters of the six bottled Shiraz wines were measured as detailed in Table S1.

All wines had very low sugar content (from <0.8 g/L to 1.5 g/L) with a percentage of ethanol between 13.28% to 16.31%, typical of wines from the south of France. Volatile

acidity levels ranged from 0.42 to 0.80 g/L $H_2SO_4$, which were all lower than the authorized limit of 0.98 g $H_2SO_4$/L, according to OIV resolutions. Since free $SO_2$ was low for some wines (8 to 29 mg/L), the levels were then adjusted for all wines to 30 mg/L. All levels of malic acid were present in lower concentrations to 0.3 g/L, meaning that malolactic fermentation was completed in all wines. Copper concentrations ranged from <1.0 to 3.1 mg/L, so there was no risk of copper breakage since all levels were inferior to 10 mg/L.

Concerning VSC, seven of these (DMS, EtSH, DES, SMTA, DMDS, ETA, DEDS) and DMSP were measured in the six wines right after bottling and after three months of accelerated bottle ageing (Table S1). Right after bottling, four out of seven VSC analyzed showed concentrations above their detection thresholds, and DES, DMDS and DEDS were not detected through GC-MS/MS in any sample. For the seven detected compounds, $H_2S$ was the major one in five wines, except for LR1 with DMS as major compounds. At this step, wine closures had no impact on VSC presence after bottling, confirming that bottling under inert gas allowed elimination of any experimental bias.

### 3.2.2. Final Characterization of Wines

#### Evolution of VSC

All the tendencies for all VSC were detailed in Table S1 and showed an overall increase during accelerated ageing of Shiraz wines. Among all VSC, only DMS showed a significant increase after accelerated wine ageing and became the major volatile sulfur compound in all wines, regardless of wine closure. Consequently, only DMS and DMSP were detailed thereafter.

#### Evolution of DMS

Under our conditions, DMS concentrations after bottling ranged from 4.1 μg/L to 315.0 ± 20.0 μg/L and rose from 60 ± 1.0 μg/L to 770 ± 88.0 μg/L after three months of accelerated bottle ageing. For each wine sample, there was a significant increase in DMS content after three months of accelerated ageing (Figure 1), ranging from 67% to 822%.

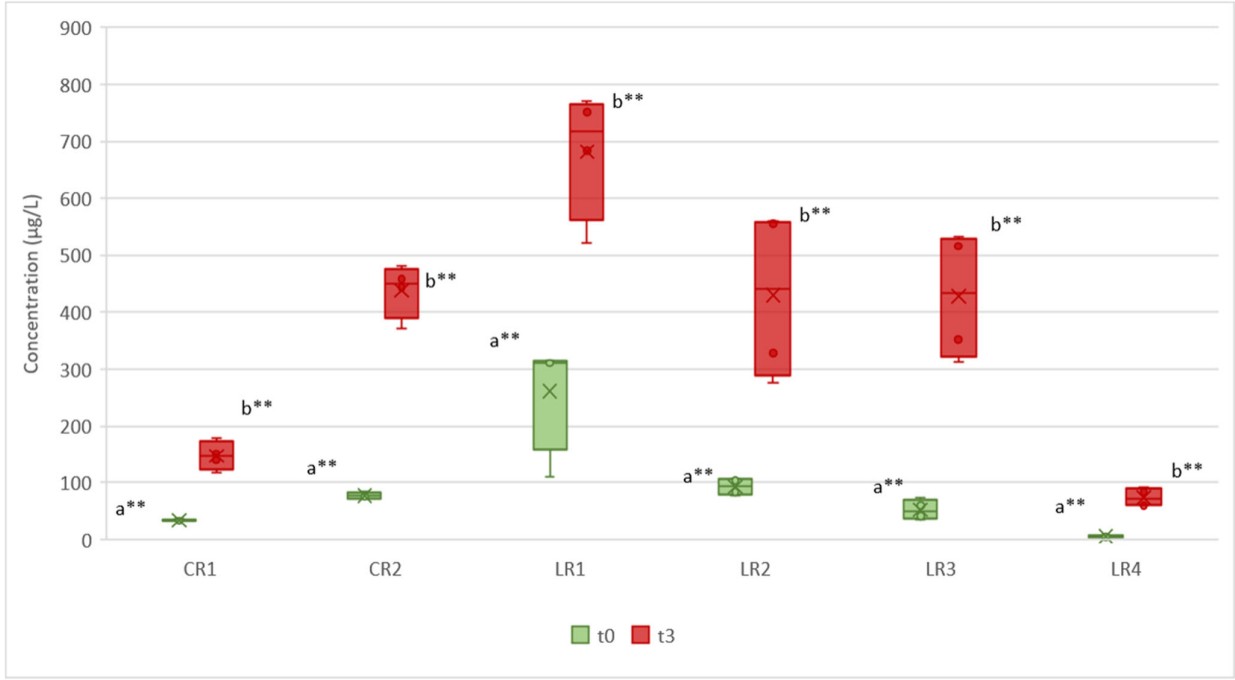

**Figure 1.** Evolution of the DMS concentration (μg/L) between bottling (t0) and after three months of accelerated bottle ageing (t3) for six different wine samples (CR1, CR2, LR1, LR2, LR3 and LR4) (a and b: a different letter means there is a significant difference between the two groups; **: significant difference at 5%).

When we take a look at the closure effect, we observed a significantly higher production of DMS with the tightest cap compared to the most permeable one (closure 1 vs. closure 4) for five out of the six studied wines (Figure 2). It is important to mention that DMS production was higher for closure 4 compared to closure 1 only for LR2. For all other comparisons between two closures, the ANOVA analysis did not show any systematic difference.

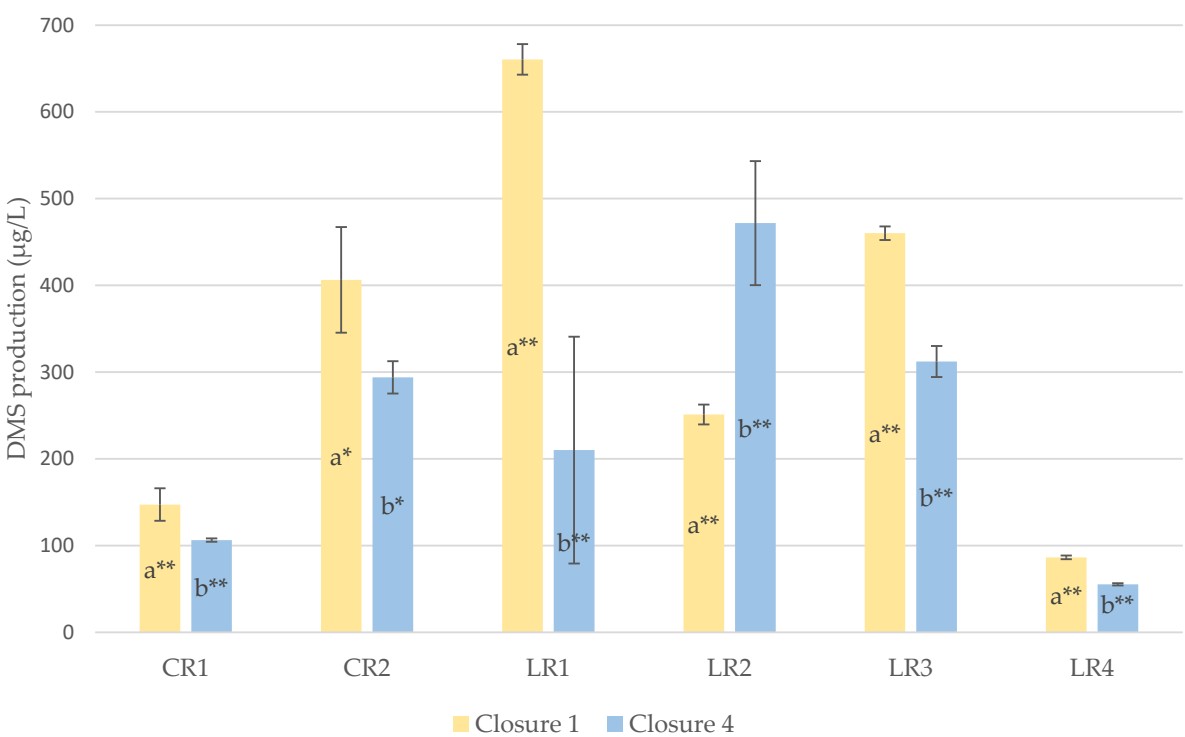

**Figure 2.** DMS formation after three months of accelerated bottle ageing depending on closure 1 (OTR = 0.19 mg $O_2$/year) and closure 4 (OTR = 1.79 mg $O_2$/year) for each wine sample (a and b: a different letter means there is a significant difference between the two groups; **: significant difference at 5%; *: significant difference at 10%).

Evolution of DMSP

Dimethylsulfide potential is an analytical indicator that corresponds to the cleavage, under alkaline treatment, of all the (known or unknown) precursors into DMS, in a specific sample, at 100 °C for 1 h [6]. This indicator has already been used to characterize the aromatic potential of grapes [6] or to investigate the water deficit effect of Grenache grapes [26]. Even if this indicator is under-used by the scientific community, it allows for observation of the macroscopic phenomemon in comparison to targeted approaches focused on SMM only.

After bottling, DMSP levels in Shiraz wines ranged from 119 µg/L to 958 µg/L and decreased from 91.7 ± 0.2 µg/L to 534 ± 36 µg/L after three months of accelerated bottle ageing. In other words, DMSP concentrations decreased significantly, from 13% to 57%, during accelerated ageing.

To go into further detail, we investigated the closure effect on DMSP degradation. To this end, an ANOVA analysis was set up to check the behavior of wine closures towards DMSP degradation. We observed significantly less DMSP degradation between the tightest closure (closure 1) and the most permeable one (closure 4) for four out of the six wines (LR1, LR2, LR4 and CR2) (Figure 3).

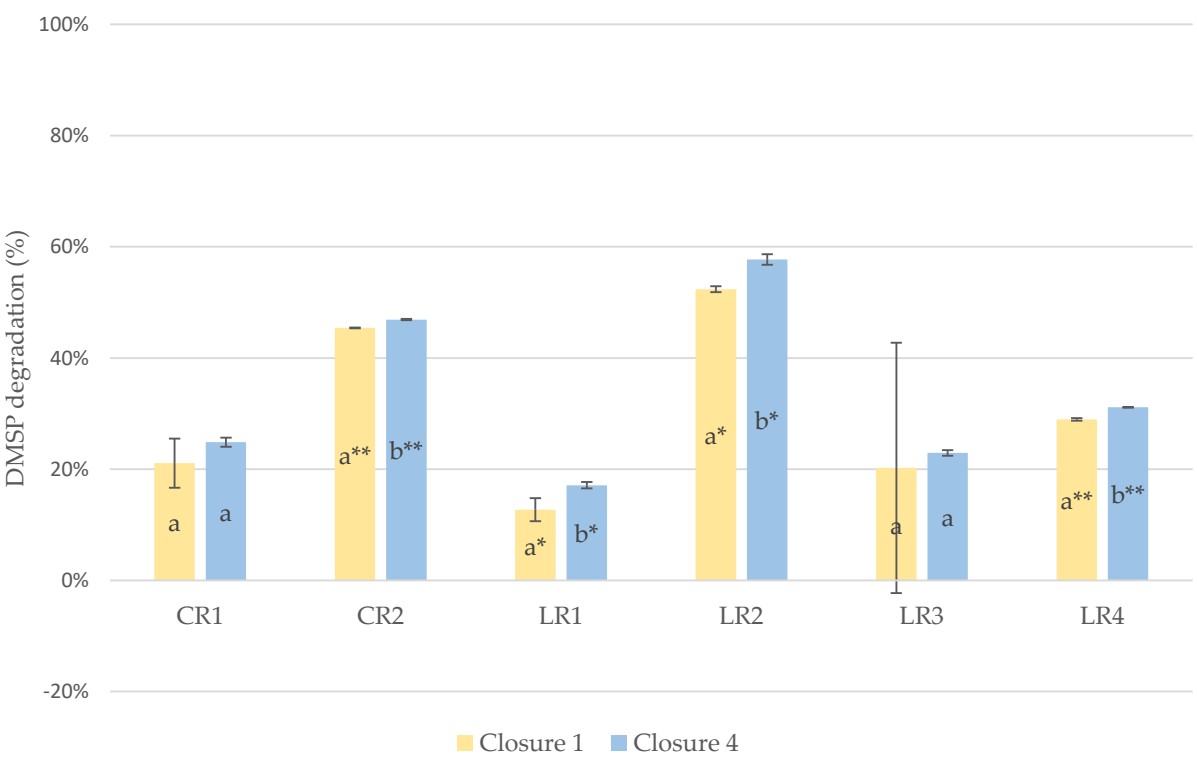

**Figure 3.** Percentage of DMSP degradation after three months of accelerated bottle ageing depending on closure 1 (OTR = 0.19 mg $O_2$/year) and closure 4 (OTR = 1.79 mg $O_2$/year) for each wine sample (a and b: a different letter means there is a significant difference between the two groups; **: significant difference at 5%; *: significant difference at 10%).

## 4. Discussion

### 4.1. Evolution of DMS

After accelerated wine ageing, the DMS levels were either within the same range as other works [13] or higher compared to data found in previous studies [6,27]. Nevertheless, it is difficult to compare these different works because DMS biosynthesis in wine has multiple origins, originating in a major part from *S*-methylmethionine in grapes and, to a lesser extent, from yeast sulfur metabolism during alcoholic fermentation. Indeed, the grape variety, yeast strain and winemaking protocols are often very different in reported works, rendering it difficult to compare absolute concentrations of DMS. Consequently, only the overall evolution during ageing can really be compared and discussed in regard to the literature.

As mentioned in the results, we observed lower levels of released DMS with the most permeable closure (Figure 2). This could be explained by DMS losses through wine closure or by a scalping phenomenon. Indeed, Silva and co-workers showed that closures could trap some VSC and that cap nature could strongly modulate the quantity of trapped molecules [17]. According to their study, compared to a control sample, the levels of DMS remained unaltered in model wine bottled with screw caps and synthetic closures, whereas the DMS concentrations decreased after 25 days for samples bottled with natural cork closures. No data on micro-agglomerated closures were found for DMS. To explain our observations, further studies would be required to quantify the DMS proportion that could be scalped into our wine closures.

From a sensory point of view, an accumulation of DMS in wine could be interesting since it is able to enhance fruity notes, especially in red wines [28]. So, controlling DMS release can be of great interest to winemakers by considering a closure with low permeability, for instance.

*4.2. Evolution of DMSP*

From a general point of view, the DMSP levels observed in this study were much higher compared to those found in 15 Shiraz and 14 Grenache wines (from 8.6 to 97.1 µg/L) in the literature [6,13]. Another study also reported DMSP levels from 30 to 100 µg/L in Grenache wines [26], which is quite a bit lower than our concentrations.

The decrease in DMSP content over accelerated wine ageing is consistent with the literature where DMSP levels decreased during bottle ageing [13]. If we take into account the regression model of DMS release from DMSP (% of released DMS = $-0.0012x^2 + 0.0594x + 0.0725$ with x = age of wine, [29]), it would appear that our accelerated aging conditions were equivalent to 14 years of standard aging with respect to DMSP degradation.

In light of the results that showed a lower DMSP degradation with the tightest closure (closure 1) and a higher DMSP degradation with the most permeable one (closure 4) (Figure 3), we raised the hypothesis that oxygen ingress through the closure catalyzed DMSP degradation into the corresponding DMS. During ageing, DMS originates in large part from the hydrolysis of SMM [12] and more generally from DMSP degradation [13]. Until now, only the acidity of the wine matrix was deemed to be involved in this degradation. These results highlight the fact that oxygen could be involved in DMSP degradation, constituting, to our knowledge, the first report of this phenomenon.

## 5. Conclusions

Under accelerated ageing conditions, DMS is the only VSC that showed a significant increase after ageing, becoming the major VSC in all six wines. It is also interesting to notice that the other VSC were generally increasing without any closure effect being related to this trend.

DMS had a significantly higher concentration in wines with the closure that had the lowest OTR. Since DMS participates in the quality of the wine, an impermeable closure would be recommended to preserve DMS. Nevertheless, some compromise might be necessary to avoid any formation of reduced off-flavor due to an accumulation of VSC. These results supplement the knowledge on flavor scalping by wine closures, by adding the study on microagglomerated closures. For the first time, closure permeability and, indirectly, oxygen level could play a role in the DMSP decrease.

From a technical point of view, closure is the last technological step that could have an impact on the DMS/DMSP equilibrium in long-keeping wines. For wines with high levels of DMS, closures with a very low permeability seem to be recommended to preserve DMS. On the contrary, for wines with a low concentration of DMS but a high DMSP level, a closure with a higher permeability would be advised to allow the production of DMS during ageing.

**Supplementary Materials:** The following supporting information can be downloaded at: https://www.mdpi.com/article/10.3390/beverages9010015/s1, Table S1: Mean concentration and standard deviation of volatile sulfur compounds in Shiraz wines according to the type of closure. Table S2: Retention time (RT) and detection conditions by tandem mass spectrometry.

**Author Contributions:** Conceptualization, A.R., C.L. and R.D.L.B.; validation, R.D.L.B. and A.R.; formal analysis, R.D.L.B., V.N. and T.G.; investigation, R.D.L.B.; resources, C.L., N.G. and D.T.; writing—original draft preparation, R.D.L.B.; writing—review and editing, A.R., N.S. and R.D.L.B.; supervision, A.R.; project administration, N.S. and A.R.; funding acquisition, A.R. All authors have read and agreed to the published version of the manuscript.

**Funding:** This research was funded by DIAM BOUCHAGE, Espace Tech Ulrich, 66400 Céret, France.

**Data Availability Statement:** Not applicable.

**Acknowledgments:** We acknowledged Diam bouchage for funding this project. Winemakers from Languedoc-Roussillon and Côtes-du-Rhône are thanked for allowing sampling in their cellars. Vivelys is acknowledged for its help in wine bottling.

**Conflicts of Interest:** The authors declare no conflict of interest. The funders had no role in the design of the study; in the collection, analyses, or interpretation of data; in the writing of the manuscript; or in the decision to publish the results.

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
