# Peer review of "Wine Cork Closures Impacts on Dimethyl Sulfide (DMS) and Precursors (DMSP) Equilibrium of Different Shiraz Wines during Accelerated Bottle Ageing"

_beverages, doi:10.3390/beverages9010015_

Round 1

Reviewer 1 Report (New Reviewer)

The article treats the effect of cork closure on the equilibrium of DMS-DMSP in Shiraz wines. This question was not yet studied in the literature and can be interesting to help winemakers to choose the adequate closure. Authors tested four micro agglomerated cork closure with different permeability, which is an important characteristic.

The research design is good, but the presentation of the results and the discussion have to be improved. If sensory analyze was did, it would be very interesting to present their results.

In generally the writing, the style and editing of the manuscript must be improved (yellow underline, revision corrections, figur annotation).

Some recommendation:

Line 99-102     Not necessary to write the values of OTR when you noted it in the table. It would be easier to read. Please define OIR in the text if you give values in the table. 

Line 136          Please correct : MgSO4 x 7H2O

Line 145          Please cite the name of the author [22].

Line 158          Same as Line 145

2.4.3.               It would be interesting to have a short description of the method of DMSP measurement here.

3.1                   This part is too long compared to the other part of Results. Table 3 gives the information important, it is not necessary to detail the validation for this method, it is not a new method, it was already published.

Line 245 and Table S1:  Could you explain why it is only 1 value/wine for DMSP right after bottling? This is somehow contradictory with the 4 value of DMS for the same state (t0).

3.2.2.1.            It would be interesting to give more details about VSC evolution even if it is not a significant effect of closure. For example, MeHS increases significantly after 3 months. You have an explication why?

Line 273-277   Does the Table 4 deleted or not. If it is, in this case you should change the sentence in Line 259. 

Figure 1.          Please correct the figure! Put the letter a** and b** on the right place!

Line 279-281   Please complete with the description of x axis. 

Line 290          Formulation of the sentence is not clear, please change.

Figure 2           It would be interesting to present results for all the 4 closures. Perhaps it could be give also an explication for the special results found for L2.

Line 336          If you use only 1 value/wine modality for DMSP at t0, please explain how you calculate the decrease of DMSP and why?

Figures 3         Same proposition as for Figure 2 

Line 367          DMS/PDMS or DMS/DMSP ?

Author Response

Thank you for your comments. Please find the attached pdf for the replies.

Reviewer 2 Report (New Reviewer)

The authors present an interesting study concerning the evolution of DMS/DMSP equilibrium in wine also investigating the effect of different closure permeabilities.

The purpose of the research is clear, the reported results are adequately discussed and the paper is well argued. The cited literature is adequate. Thus, I recommend to publish the paper in Beverages after minor revision. I have few comments/suggestions:

-       In my opinion, authors should better describe the reason for which Shiraz has been selected for investigating the evolution of DMS/DMSP equilibrium during ageing.

-       What about the use of a white wine?

-       In section 2.3.3. “Accelerated ageing mode” please indicate the bottle type.

-       Were samples stored in the dark or under light exposure?

Author Response

This manuscript is a resubmission of an earlier submission. The following is a list of the peer review reports and author responses from that submission.

Round 1

Reviewer 1 Report

The manuscript deals with the dimethyl sulfide potential in wine, along with the changes of its concentration during accelerated storage. Moreover, the effect of closures with different oxygen transfer rates on DMS concentration was also investigated. Although the proposed paper is interesting and scientifically important, there are serious lacks and should be rejected.

Introduction- used literature should be more up-to-date.

Materials and methods

Wine closures section- The dimensions of the closures should be state. Moreover, there is no need to write OTR in text and in Table. Generally, certain data should be stated only once. Also, O2 in Table should be written as O2

Wine bottling- each wine was bottled with all four closures…. 66 bottles × 4 closures is 264 bottles in total, not 232 as stated in text.

Accelerated ageing mode – for what did you used remaining bottles? At the beginning, you had 1600 bottles. 72 of them were analyzed at the beginning, and 72 after 3 months of accelerated ageing.

Generally, this section (Materials) should be improved and written more clearly.

Sample preparation- this procedure should be conducted in inert conditions, to avoid oxidation of sulfur compounds. For example, as presented in paper Slaghenaufi et al. (2017) [reference 23]

Results

Table 4- PDMS should be rewritten- DMSP; LR4 results- only one bottle analyzed? Among 1600 bottles in start?

Figure 1. – these results are the same as the ones presented in Table 1. As stated previously, the same data should not be presented in two forms.

Figure 3.- are the data of LR3 sample correct, due the extreme standard deviation?

Conclusion/Abstract- the claim that ‘DMS escapes through the cork closure’ should be revised. Moreover, the terminology of tightness in terms of closure also should be revised. OTR is dependent on the permeability of the closures to oxygen- it does not mean that closures that have lower OTR are more tighten, or the ones that have higher OTR are loosen. For that reason, the dimensions of the closures should be stated in Material section, as well.

References- the more up-to-date literature should be used.

Reviewer 2 Report

The study entitled “Wine cork closures affects the dimethyl sulfide – dimethyl sulfide potential equilibrium during accelerated bottle ageing of Shiraz wines” by De la Burgade et al. is original scientific research. Unfortunately, despite the subject that is interesting, this paper faces some limitations considering the implementation of the study and obtained results. Hance, to my opinion, this manuscript should not be published in Beverages journal. Some explanations are given below.

Authors put great focus to the analytical validation of the method, and jet in the Table 4 they present extrapolated results. Also, for the samples LR4 in the same table, there are some missing data. If the part of data for one sample is missing, then this sample should be excluded from the study, particularly since the results obtained for DMS for LR4 sample and all 4 closures significantly differ from all the other wines. In addition, Table S1 is missing, also table S2 is mentioned in text but missing.

Part of material and methods explaining number of samples used in the study should be double-checked since some numbers does not match the description given in text.

In addition, another limitation of this study is that authors did not put enough focus to the changes of oxygen. Namely, authors did not measure changes of the oxygen concentration in the bottle during accelerated aging. This date would be very valuable, particularly since the authors included different wines in the study and evolution of each wine is not dependent just of OTR and OIR information. Also, it would be valuable if authors performed sensory analysis of wine.

Conclusion in the abstract is that DMS could escape through the closure is also unclear because measurements conducted in this work does not prove that. It is very hypothetic conclusion.

In addition, DMS does not come from “potential”. Formulation of the sentence is wrong. DMS potential can increase and decrease but term “degradation of potential” should be replaced with more suitable term.

Reviewer 3 Report

Please refer the attached file
